# A mixed methods evaluation of medication reconciliation in the primary care setting

**Michael R. Gionfriddo**[1¤]*, **Vanessa Duboski**[1], **Allison Middernacht**[2], **Melissa S. Kern**[1], **Jove Graham**[1], **Eric A. Wright**[1]

1 Center for Pharmacy Innovation and Outcomes, Geisinger, Danville, PA, United States of America,
2 Wilkes University School of Pharmacy, Wilkes-Barre, PA, United States of America

¤ Current address: Division of Pharmaceutical, Administrative and Social Sciences, Duquesne University School of Pharmacy, Pittsburgh, PA, United States of America
* gionfriddom@duq.edu

**Data Availability Statement:** All relevant data are within the manuscript and its Supporting Information files.

## Abstract

### Objectives

To understand the extent to which behaviors consistent with high quality medication reconciliation occurred in primary care settings and explore barriers to high quality medication reconciliation.

### Design

Fully mixed sequential equal status design including ethnographic observations, semi-structured interviews, and surveys.

### Setting

Primary care practices within an integrated healthcare delivery system in the United States.

### Participants

We conducted 170 observations of patient encounters across 15 primary care clinics, 48 semi-structured interviews with staff, and 10 semi-structured interviews with patients. We also sent out surveys to 2,541 eligible staff with 616 responses (24% response rate) and to 5,132 eligible patients with 577 responses (11% response rate).

### Results

Inconsistency emerged as a major barrier to effective medication reconciliation. This inconsistency was present across a variety of factors such as the lack of standardized workflows for conducting medication reconciliation, a lack of knowledge about medication and the process of medication reconciliation, varying levels of importance ascribed to medication reconciliation, and inadequate integration of medication reconciliation into clinical workflows. Findings were generally consistent across all data collection methods.

**Funding:** This work was funded by the Geisinger Health Plan through the Geisinger Clinic Quality Pilot Fund program for fiscal years 2020 and 2021. The funders had no role in study design, data collection and analysis, decision to publish, or preparation of the manuscript.

**Competing interests:** This work was funded by the Geisinger Health Plan through the Geisinger Clinic Quality Pilot Fund program for fiscal years 2020 and 2021. The sponsor had no role in the design, conduct, analysis, or reporting of the study. MRG and EW report receiving funding in past 3 years from Merck Sharp & Dohme Corp to develop a tool to improve medication reconciliation. All other authors declare no relevant interests. This does not alter our adherence to PLOS ONE policies on sharing data and materials.

## Conclusion

We have identified several barriers which impact the process of medication reconciliation in primary care settings. Our key finding is that the process of medication reconciliation is plagued by inconsistencies which contribute to inaccurate medication lists. These inconsistencies can be broken down into several categories (standardization, knowledge, importance, and inadequate integration) which can be targets for future studies and interventions.

## Introduction

Inaccurate or incomplete medication reconciliation results in over 100,000 preventable hospital admissions and over $1 billion in excess healthcare costs yearly [1–4]. Medication reconciliation has been defined by an international consensus group as:

> The process of creating the most accurate list possible of all medications a patient is taking and comparing that list against the prescriber's orders. In addition, the patient's allergies, history of side effects from medications and medication aids are listed with the goal of providing correct medication to the patient at all transition points within the health care system [5].

The process of medication reconciliation does not always occur and when it does, the effectiveness is uncertain [6, 7]. Studies, including those completed in primary care environments, have found that there are often numerous discrepancies between the list held by the patient and that held by the health system [7–13]. While some of these discrepancies will naturally arise as patients' conditions and contexts change, often there are barriers which prevent the effective completion of medication reconciliation. This may result in discrepancies persisting on patient medication lists for years. These barriers include lack of time, challenges associated with the electronic health record (EHR) and communication across settings of care, patient lack of knowledge of their medications, and the lack of standardized workflows [14, 15]. Few studies, however, assess these barriers from both the patient and clinician perspective. Additionally, there is limited information on the extent through which behaviors consistent with a high-quality medication reconciliation occur in primary care.

To understand the process of and barriers to high quality medication reconciliation, we conducted a mixed methods evaluation as part of a quality improvement project to improve medication reconciliation in our primary care practices. Specifically, this study aims to answer the following research questions:

1. To what extent are behaviors indicative of a best possible medication history (BPMH) for medication reconciliation adhered to in a primary care setting?

2. What are the barriers which inhibit the accurate completion of medication reconciliation? Do these barriers differ between patients and staff or between staff with and without prescribing authority?

3. What changes do patients and staff think can be made to improve the process of medication reconciliation in primary care?

## Methods

### Setting

Geisinger is an integrated healthcare delivery system that serves part of Pennsylvania, delivering care to approximately 4.2 million residents. In addition to Geisinger's seven hospital campuses, the providers serve a network of 130 primary and specialty clinic sites, of which over 40 are community-based primary care clinics.

### Study design

We conducted a fully mixed sequential equal status mixed methods [16] evaluation of medication reconciliation within primary care clinics combining ethnographic observations, semi-structured interviews, and surveys. The observations occurred first and informed both survey and interview guide development. For example, we noted in the observations that the adherence to best practices for conducting a BPMH were variable. To better understand this variability, we asked questions in the interview and surveys such as whether the staff had been trained in conducting medication reconciliation/BPMH. Similarly, in the observations we noted that over the counter (OTC)/non-prescription medications were asked about infrequently so we specifically asked about this during the interviews. Following the observations, we performed staff interviews from which we identified themes that led us to create targeted questions to assess in the staff survey to understand the applicability of the themes to the broader population of staff at our institution. For example, during the interviews we noted that time emerged as a barrier and was thus included on the staff survey. The patient survey, which was informed by the observations and interviews with staff was then used to recruit patients to interview. The findings from each method are mixed in the results based on thematic categorization and act as a form of triangulation.

The work presented in this manuscript was approved by the Geisinger Institutional Review Board. The observations were considered quality improvement (IRB#2019–0561), while the surveys and interviews were deemed exempt (IRB# 2019–0868).

### Focused ethnography

To address the extent to which behaviors consistent with a BPMH are adhered to in practice (Research Question 1), we conducted ethnographic observations of primary care clinics between June and September of 2019. The clinics, staff, and encounters observed were chosen through convenience sampling in consultation with clinic leadership. On pre-specified days, observers (AM and VD) were assigned to a staff member (nurse or medical assistant) or rotated between available staff at clinic discretion. The observer would observe encounters at staff discretion. The goal was for the observer to follow a patient throughout the course of their encounter with both the nurse or medical assistant and primary care provider. This meant observing a medication history completed by a nurse and a medication reconciliation completed by a provider whenever feasible. Patient encounters were not observed if the nature of the visit was sensitive or either the staff or patient felt uncomfortable or preferred not to have an observer present during the encounter.

The observers, a pharmacy student on a summer internship (AM) and a project coordinator (VD), underwent a brief training with an experienced qualitative researcher (MG) and used a structured observation guide (Appendix A in S1 File) that focused on behaviors which facilitate a BPMH. These behaviors were chosen based on a review of the literature and established best practices [17–19]. In addition to the structured observation guide, observers

documented the time to complete a medication history and entered free text reflections on the observation guide which were discussed with the research team.

Data were summarized using descriptive statistics with Microsoft Excel® (Microsoft, Redmond, WA). We also reviewed quantitative data for relationships between variables (e.g. between staff type and adherence to behaviors). This data and any qualitative reflections by the observers informed the subsequent development of semi-structured interview guides and surveys.

## Interviews

We conducted interviews with a variety of stakeholders to understand their perspectives on, and experience with, medication reconciliation including perceived barriers and suggestions for improvement (Research Questions 2 and 3). Stakeholders included physicians, physician assistants, nurse practitioners, clinical pharmacists, informaticians, nurses, case managers, clinic managers, and patients. All stakeholders were recruited via purposive sampling. The staff were initially recruited via known contacts from either prior observations or from participation in related historical or ongoing quality improvement initiatives. We recruited additional staff through snowball sampling; as part of the interview, staff were asked to identify others who could contribute to our understanding of medication reconciliation. Patients were initially recruited through lists generated from recent clinic visits; after several interviews, however, it was determined that a more purposive approach was necessary and additional patient interviews were scheduled with patients who agreed to a follow-up interview through the survey (see below for additional details on the survey). Patients were chosen using a maximum variation approach based on their survey answers.

We conducted all interviews using semi-structured interview guides which were tailored to stakeholder type. One set of three interview guides were developed for staff (Appendix B in S1 File), and another guide was developed for patients (Appendix C in S1 File). The guides were informed by findings from the observations as well as the research teams' healthcare experience and their involvement in related quality improvement initiatives. The individual semi-structured interviews were conducted by either an experienced qualitative researcher (MG) or a trained project coordinator (VD) and lasted up to an hour. Staff did not receive compensation for participating, while patients received a $10 gift card.

We audio recorded and transcribed interviews verbatim using a third-party service offered by our institution. Two members of the research team (MG and VD) reviewed the transcripts for accuracy. For staff, the transcript was returned to the stakeholder for review and comment (no stakeholders provided any corrections or additional comments). Several members of the research team reviewed the transcripts and documented their reflections on an ongoing basis to facilitate data interpretation and thematic analysis [20]. Two members of the research team (MG and VD) then created a codebook based on initial interviews and independently, inductively coded the transcripts using ATLAS.ti 8 Windows (ATLAS.ti Scientific Software Development GmbH, Berlin, Germany) while modifying the codebook as necessary and communicating any changes to each other. After coding was complete, the coders presented their findings to the larger research team who discussed and triangulated the findings based on their experience and reflections. Final themes were collaboratively developed by the research team. Interviews were stopped once we achieved data saturation based on the consistency of themes across these interviews.

## Surveys

To gather additional data to understand the barriers to medication reconciliation and suggestions for improvement (Research Questions 2 and 3), we conducted surveys of both patients

and staff. We conducted staff surveys to explore the credibility of our interview findings, while patient surveys helped identify patients for semi-structured interviews (see above). Patient stakeholders were identified from our EHR. We queried our EHR for all patients at least 18 years of age, seen at one of the primary care clinics at Geisinger for a visit with their primary care provider in the past three months prior to the administration of the survey, had at least one medication on their profile, and had an e-mail address on file. The survey was informed by the observations, staff interviews, and existing literature (Appendix D in S1 File) [15, 21]. The survey was 19 questions and based on pilot testing, completion should have taken no more than 15 minutes. We administered the survey using Research Electronic Data Capture, or REDCap (REDCap, Nashville, Tennessee), a secure web application for building and managing online surveys and databases [22, 23]. We sent out two reminder e-mails at weekly intervals. Patients were not offered any compensation for completion but were offered the opportunity to discuss their responses and experiences with a member of the research team in a follow-up semi-structured interview (described above).

The staff survey was developed in a similar manner based on findings from the observations, staff interviews, and existing literature (Appendix E in S1 File) [15, 21]. Working with clinical leadership, we identified staff working in primary care and invited them to participate in the survey via e-mail. This survey had two preliminary questions to first assess staff eligibility (working in an ambulatory setting and regularly completing any component of medication reconciliation). If "yes" was answered to these two questions, the staff member was then routed to the remaining 28 survey questions. Based on pilot testing, completion should have taken no more than 20 minutes. This survey was also administered using REDCap (REDCap, Nashville, Tennessee) [22, 23]. We sent out two reminder e-mails at weekly intervals, and staff received no compensation for participation.

For both patients and staff, we analyzed the survey responses using descriptive statistics and explored patterns in the data (e.g. prescriber vs. non-prescriber). Prescribers were defined as pharmacists, physicians (both primary care and specialty), physician assistants, and certified registered nurse practitioners; non-prescribers included: nurses, case managers, community health associates, and medical assistants. For the patient survey, we also conducted a non-response analysis examining the impact of various factors (e.g. age, sex, race, number of medications, etc.) on responder status (i.e. responder vs. non-responder). Comparisons of continuous data were conducted using t-tests, while comparisons of categorical data were conducted using chi-square or Fisher's exact tests as appropriate. All statistical analyses were done with SAS (SAS 9.4, Cary, NC) or R (The R Group, Vienna, Austria) and p-values $<0.05$ were considered significant; no adjustment was made for multiple comparisons.

## Results

We conducted 170 observations of patient encounters across 15 primary care clinics, 48 semi-structured interviews with staff, and 10 semi-structured interviews with patients (Appendices F and G in S1 File). We also sent out surveys to 2,541 eligible staff with 616 responses (24% response rate) and to 5,132 eligible patients with 577 responses (11% response rate). Complete flow diagrams can be found in Appendices H and I in S1 File. Respondents to the staff survey had an average age of 43 years (standard deviation (SD) of 12), were mostly female (82%), and had been in their position an average of 10 years (SD 10). Respondents practiced in a variety of settings and had a variety of roles (Table 1).

Respondents to the patient survey had an average age of 60 years (SD 18), 56% were female, the majority were White (96%), had commercial insurance (76%), and utilized Geisinger's electronic patient portal (90%) (Table 2).

**Table 1. Staff survey demographics.**

| Question | Responses | Respondents, N (%) (n = 616) |
|---|---|---|
| What is your age? | 0–24 | 20 (3%) |
| | 25–34 | 312 (51%) |
| | 45–64 | 258 (42%) |
| | 65+ | 26 (4%) |
| | Mean (SD) | 43 (12) |
| | Range | (21, 75) |
| What is your gender? | Female | 505 (82%) |
| | Male | 100 (16%) |
| | Other | 2 (<1%) |
| | Prefer not to say | 9 (1%) |
| Practice Area | Primary Care | 331 (54%) |
| | Specialty Care | 257 (42%) |
| | Other* | 28 (4%) |
| Position | Case Manager | 33 (5%) |
| | Community Health Assistant | 15 (2%) |
| | Licensed Practical Nurse | 181 (29%) |
| | Medical Assistant | 53 (9%) |
| | Nurse Practitioner | 30 (5%) |
| | Pharmacist | 42 (7%) |
| | Physician Assistant | 67 (11%) |
| | Physician Specialist | 50 (8%) |
| | Primary Care Physician | 63 (10%) |
| | Registered Nurse | 65 (11%) |
| | Other | 17 (3%) |
| Years in position? | 0–1 | 112 (18%) |
| | 2–4 | 162 (26%) |
| | 5–9 | 117 (19%) |
| | 10–14 | 81 (13%) |
| | 15–29 | 42 (7%) |
| | 20–29 | 54 (9%) |
| | 30+ | 48 (8%) |
| | Mean (SD) | 9.7 (10.3) |
| | Range | (0, 47) |

*Examples of other includes individuals working in emergency medicine, skilled nursing facilities, medication therapy management, etc.

As part of the patient survey, 294 patients agreed to be interviewed, 34 were invited, and 10 completed semi-structured interviews. A non-response analysis found that respondents were older, with higher comorbidity scores, on more medications, more likely to have and use our patient portal, and were more likely to have either Geisinger Health Plan or Medicare insurance while being less likely to have other commercial insurance. Respondents had an average of 10.4 (SD 7) medications listed in the EHR but self-reported an average of 7.0 (SD 4) medications. Table 3 summarizes our findings based on each of our initial research questions.

**Table 2. Patient survey demographics.**

| | Respondents, N (%) |
| --- | --- |
| | **(n = 577)** |
| Sex, N (%) | |
| Female | 326 (56%) |
| Male | 251 (44%) |
| Race, N (%) | |
| American Indian or Alaska Native | 0 (0%) |
| Asian | 3 (<1%) |
| Black or African American | 15 (2.6%) |
| Native Hawaiian or Pacific Islander | 2 (<1%) |
| White | 555 (96%) |
| Unknown | 2 (<1%) |
| Age, N (%) | |
| <25 | 11 (2%) |
| 25–44 | 76 (13%) |
| 45–64 | 247 (43%) |
| 65–84 | 227 (39%) |
| 85+ | 16 (3%) |
| Mean (SD) | 60 (18) |
| Charlson Comorbidity Index, N (%) | |
| 0 | 68 (12%) |
| 1–5 | 388 (67%) |
| 6–10 | 113 (20%) |
| 11+ | 8 (1%) |
| Insurance, N (%) | |
| Geisinger Health Plan | 225 (39%) |
| Medicare | 123 (21%) |
| Medicaid | 1 (<1%) |
| Other Commercial | 212 (37%) |
| Other | 13 (2%) |
| Unknown | 3 (<1%) |
| Enrolled in MyGeisinger, N (%) | |
| Yes | 537 (93%) |
| No | 40 (7%) |
| Recent User of MyGeisinger, N (%) | |
| Yes | 519 (90%) |
| No | 58 (10%) |
| Number of active meds in EHR | |
| Mean (SD) | 10.4 (7.0) |
| Median (IQR) | 9 (5, 14) |
| Range | (1, 47) |
| Patient reported medication use | |
| Mean (SD) | 7.0 (4) |
| Median (IQR) | 6 (4–10) |
| Range | 0–26 |
| On average, how often do you visit a healthcare provider? | |
| More than once a week | 5 (1%) |
| Once a week | 7 (1%) |
| A few times a month | 27 (5%) |
| Once a month | 47 (8%) |

(*Continued*)

**Table 2.** (Continued)

| | Respondents, N (%) |
|---|---|
| | **(n = 577)** |
| Every few months | 378 (66%) |
| Once a year | 101 (18%) |
| Less than once a year | 12 (2%) |

**Table 3. Main findings.**

| Research Question | Main finding |
|---|---|
| 1. To what extent are behaviors indicative of a best possible medication history (BPMH) for medication reconciliation adhered to in a primary care setting? | Average adherence to behaviors consistent with a BPMH was 53% and ranged from 33% to 82%. Adherence to individual behaviors were also variable ranging from 2% to 100%. Notable findings include: |
| | • OTC medications were asked about by 36% of observed staff |
| | • Medication name was asked 99% of the time while dose was confirmed only 41% of the time |
| | • Staff asked if the patient was taking any new medications 17% of the time |
| 2. What are the barriers which inhibit the accurate completion of medication reconciliation? Do these barriers differ between patients and staff or between staff with and without prescribing authority? | We found that inconsistency was a major driver of poor medication reconciliation and identified several barriers including: |
| | • **Lack of a standardized workflow**. This contributed to a lack of clarity around who is responsible for medication reconciliation and lack of comfort around conducting medication reconciliation. |
| | • **Lack of knowledge**. This included lack of knowledge about how to conduct proper medication reconciliation as well as lack of knowledge about medications (for both patients and staff) |
| | • **Variable importance of medication**. While both staff and patients recognized the importance of having an accurate medication list, importance varied based on the type of medication being addressed with prescription medications generally viewed as more important than non-prescription medications. |
| | • **Inadequate integration into clinical workflows**. Staff noted that collecting medication information and entering medications into the electronic health record was difficult within the current workflows especially for non-prescription medications and existing workflows did not set aside adequate time to properly conduct medication reconciliation. |
| | • Certain barriers affecting the completion of medication reconciliation did differ between prescribers and non-prescribers (e.g., time), while others (e.g., knowledge) did not. |
| 3. What changes do patients and staff think can be made to improve the process of medication reconciliation in primary care? | Based on data from both patients and staff we identified several opportunities for improvement including: |
| | • Staff training and patient education |
| | • Reminders for patients to bring in medications lists/bottles and for staff to ask about all medications |
| | • Workflow redesign and standardization (including guidelines) |
| | • EHR redesign |
| | • Increased time for medication reconciliation or designated visits to conduct medication reconciliation. |

## Inconsistency as a driver of poor medication reconciliation

Inconsistency emerged as a major barrier to effective medication reconciliation. This inconsistency was present across a variety of factors such as the lack of standardized workflows for conducting medication reconciliation, a lack of knowledge about medication and the process of medication reconciliation, varying levels of importance ascribed to medication reconciliation, and inadequate integration of medication reconciliation into clinical workflows.

**Lack of standardized workflow.** We identified a lack of a standardized workflow during our observations. We found that adherence to behaviors consistent with a BPMH ranged from 2% to 100% with an average adherence of 53% (Table 4).

Staff were aware of this lack of a standardized workflow with only 38% feeling that a standardized process was currently in place (Table 5) with non-prescribers more likely to agree

**Table 4. Adherence rates of BPMH checklist items from clinic observations.**

| BPMH Item | Overall Adherence |
|---|---|
| Access patient's medication list | 100% |
| Review medication list | 100% |
| Check mark as reviewed | 100% |
| Verify current medications | 99.4% |
| Ask about using medications as prescribed | 99.4% |
| Ask about SIG | 99.4% |
| SIG: name | 99.4% |
| Verification of patient identification | 97.6% |
| Clarification of pharmacy | 94.7% |
| Make note as not taking, if applicable | 94.7% |
| Clarification of allergies | 91.2% |
| Add new medication, if applicable | 85.7% |
| D/C medications, if applicable | 84.9% |
| Change SIG, if applicable | 84.6% |
| Make note to physician, if applicable | 73.2% |
| Reason discussed for held medication, if applicable | 71.4% |
| Delete duplicates, if applicable | 71.4% |
| SIG: frequency | 49.4% |
| Document side effects, if applicable | 47.4% |
| SIG: last taken | 47.0% |
| Ask about D/C medication | 43.3% |
| SIG: dose | 40.9% |
| Ask about OTC medications | 35.9% |
| Ask about other concerns | 27.1% |
| Ask about other prescription medications not listed | 25.9% |
| Ask about PRN medications | 24.1% |
| Ask about new medications | 16.5% |
| SIG: indication | 13.4% |
| Ask about side effects | 12.2% |
| Use go reconcile button | 7.6% |
| Ask about adherence | 5.3% |
| Ask about held medications | 4.9% |
| SIG: dosage form | 3.0% |
| SIG: route | 1.8% |
| **Overall Adherence** | 52.8% (SD = 8.4%, Range 33.3%-81.8%) |

**Table 5. Staff survey responses.**

| Question | Responses | Total |
|---|---|---|
| **Attitudes about medication reconciliation (n = 616)** | | |
| How important is medication reconciliation in the patient care process? | Not Important | 0 (0%) |
| | Somewhat important | 14 (2%) |
| | Important | 101 (16%) |
| | Very important | 501 (81%) |
| How important is it that medication reconciliation occurs at every visit. | Not Important | 2 (<1%) |
| | Somewhat important | 41 (7%) |
| | Important | 133 (22%) |
| | Very important | 440 (71%) |
| The process of medication reconciliation is standardized across Geisinger. | Strongly Disagree | 66 (11%) |
| | Disagree | 162 (26%) |
| | Neither Agree Nor Disagree | 150 (24%) |
| | Agree | 186 (30%) |
| | Strongly Agree | 52 (8%) |
| Having a standardized process for medication reconciliation across Geisinger would be beneficial. | Strongly Disagree | 8 (1%) |
| | Disagree | 3 (<1%) |
| | Neither Agree Nor Disagree | 18 (3%) |
| | Agree | 216 (35%) |
| | Strongly Agree | 371 (60%) |
| I have a well-defined role and know what I am responsible for in the medication reconciliation process. | Strongly Disagree | 11 (2%) |
| | Disagree | 41 (7%) |
| | Neither Agree Nor Disagree | 65 (11%) |
| | Agree | 255 (41%) |
| | Strongly Agree | 244 (40%) |
| **Comfort with medication reconciliation (n = 610)** | | |
| How comfortable are you with your role in medication reconciliation? | Not Comfortable | 11 (2%) |
| | Somewhat Comfortable | 82 (13%) |
| | Comfortable | 235 (39%) |
| | Very Comfortable | 282 (46%) |
| How comfortable are you adding medications to a patient's medication list while conducting medication reconciliation? | Not Comfortable | 42 (7%) |
| | Somewhat Comfortable | 74 (12%) |
| | Comfortable | 198 (32%) |
| | Very Comfortable | 296 (49%) |
| How comfortable are you removing medications from a patient's medication list while conducting medication reconciliation? | Not Comfortable | 42 (7%) |
| | Somewhat Comfortable | 89 (15%) |
| | Comfortable | 198 (32%) |
| | Very Comfortable | 281 (46%) |
| I do not believe it is my responsibility to conduct medication reconciliation. | Strongly Disagree | 264 (43%) |
| | Disagree | 204 (33%) |
| | Neither Agree Nor Disagree | 75 (12%) |
| | Agree | 41 (7%) |
| | Strongly Agree | 26 (4%) |

(*Continued*)

**Table 5.** (Continued)

| Question | Responses | Total |
|---|---|---|
| There are unclear guidelines for what I can and cannot remove from patients' medication lists during medication reconciliation. | Strongly Disagree | 116 (19%) |
| | Disagree | 156 (26%) |
| | Neither Agree Nor Disagree | 131 (21%) |
| | Agree | 153 (25%) |
| | Strongly Agree | 54 (9%) |
| I am uncomfortable conducting medication reconciliation due to my limited knowledge of medications. | Strongly Disagree | 319 (52%) |
| | Disagree | 219 (36%) |
| | Neither Agree Nor Disagree | 41 (7%) |
| | Agree | 26 (4%) |
| | Strongly Agree | 5 (1%) |
| I am uncomfortable removing medications I did not prescribe/are not in my area of expertise, | Strongly Disagree | 162 (27%) |
| | Disagree | 150 (25%) |
| | Neither Agree Nor Disagree | 98 (16%) |
| | Agree | 149 (24%) |
| | Strongly Agree | 51 (8%) |
| **Experience with medication reconciliation (n = 599)** | | |
| While conducting medication reconciliation have you ever found an error which had the potential to cause harm to the patient? | Yes | 332 (55%) |
| | No | 268 (45%) |
| While conducting medication reconciliation have you ever identified an error which you believe did cause harm to the patient? | Yes | 81 (14%) |
| | No | 518 (86%) |
| How often do you identify errors on patients' medication lists? | Never | 14 (2%) |
| | Rarely | 268 (45%) |
| | About half the time | 232 (39%) |
| | Most of the time | 73 (12%) |
| | Always | 12 (2%) |
| After you finish conducting medication reconciliation, how confident are you that a patient's medication list is an accurate reflection of the medications they are taking? | Not Confident | 15 (3%) |
| | Somewhat Confident | 191 (32%) |
| | Confident | 300 (50%) |
| | Very Confident | 93 (16%) |
| How often do patients bring in their medication bottles from home to visits? | Never | 37 (6%) |
| | Rarely | 442 (74%) |
| | About half the time | 110 (18%) |
| | Most of the time | 9 (2%) |
| | Always | 1 (<1%) |
| Do you ask patients to bring in their medication bottles from home to visits? | Yes | 293 (49%) |
| | No | 306 (51%) |
| How often do patients bring in a medication list from home to visits? | Never | 15 (3%) |
| | Rarely | 235 (39%) |
| | About half the time | 277 (46%) |
| | Most of the time | 70 (12%) |
| | Always | 2 (<1%) |
| If patients were to bring their medication bottles or a medication list to visits, it would help me in my role of conducting medication reconciliation. | Strongly Disagree | 7 (1%) |
| | Disagree | 12 (2%) |
| | Neither Agree Nor Disagree | 58 (10%) |
| | Agree | 243 (41%) |
| | Strongly Agree | 279 (47%) |

(*Continued*)

**Table 5.** (Continued)

| Question | Responses | Total |
|---|---|---|
| **Barriers with medication reconciliation (n = 593)** | | |
| *How often do you encounter the following barriers to conducting medication reconciliation?* | | |
| Patients are not knowledgeable about their medications. | Never | 2 (<1%) |
| | Rarely | 68 (11%) |
| | About half the time | 330 (56%) |
| | Most of the time | 177 (30%) |
| | Always | 16 (3%) |
| I do not have time to conduct a thorough medication reconciliation/other tasks take priority. | Never | 115 (19%) |
| | Rarely | 216 (36%) |
| | About half the time | 133 (22%) |
| | Most of the time | 99 (17%) |
| | Always | 30 (5%) |
| Patients do not want to participate. | Never | 62 (10%) |
| | Rarely | 334 (56%) |
| | About half the time | 157 (26%) |
| | Most of the time | 37 (6%) |
| | Always | 3 (1%) |
| There are language barriers between myself and some patients. | Never | 75 (13%) |
| | Rarely | 457 (77%) |
| | About half the time | 48 (8%) |
| | Most of the time | 11 (2%) |
| | Always | 2 (<1%) |
| Patients receive healthcare outside of Geisinger. | Never | 4 (1%) |
| | Rarely | 191 (32%) |
| | About half the time | 347 (59%) |
| | Most of the time | 45 (8%) |
| | Always | 5 (1%) |
| Entering patient reported medication in the Electronic Health Record is difficult. | Never | 171 (29%) |
| | Rarely | 275 (46%) |
| | About half the time | 89 (15%) |
| | Most of the time | 38 (6%) |
| | Always | 20 (3%) |
| Other | Never | 462 (78%) |
| | Rarely | 50 (8%) |
| | About half the time | 56 (9%) |
| | Most of the time | 13 (2%) |
| | Always | 12 (2%) |
| **Training on medication reconciliation (n = 592)** | | |
| Have you ever had a formal training focusing on medication reconciliation from Geisinger? | Yes | 175 (30%) |
| | No | 417 (70%) |
| I would benefit from additional training on medication reconciliation. | Strongly Disagree | 68 (11%) |
| | Disagree | 104 (18%) |
| | Neither Agree Nor Disagree | 198 (33%) |
| | Agree | 180 (30%) |
| | Strongly Agree | 42 (7%) |

there was a standardized process (47% vs. 28%, relative risk ratio (RRR) 1.71, 95% confidence interval (CI) = 1.37 to 2.15, p<0.0001).

> *It's not consistently done at the clinics and nursing staff is supposed to do a first pass at it and then providers you know finish up or you know double check it or whatever. I don't get the sense that that's consistently happening.* (Physician/Clinic Leadership, P1)

> *I feel like I might do it one way, another nurse might do it the other way and I kind of wish that we all did the same way. . .That is important, I think. (*Nurse, P7*)*

Not only was the process of collecting the medication list inconsistent, communication about the medication list was also inconsistent and was often noted as a major barrier to medication reconciliation by staff. This occurred not only at transitions of care (e.g. hospital to home or vice-versa), but also between staff at a single site. Nurses did not have a consistent way of communicating their findings about a patient's medication use to providers. Depending on clinic and provider preference, nurses would communicate their findings in a variety of ways including: speaking with the provider directly, leaving a note on the door, documenting the changes in their nursing note in the EHR, or utilizing a note function on the medication list itself in the I.

> *It varies and I think that's one of the places where we need to do better because I'll admit that some. . .it's not always consistent where that information can come from. It could be in the nursing note. It could be on the. . .through the MedRec piece that I don't feel like that really gets communicated very well from the nurse to the physician.* (Physician/Clinic Leadership, P2)

Additionally, providers did not consistently document changes to the medication list. For example, if the patient needed to change the dose of a medication due to a side effect or a lab value this may have been done over the phone or through the patient portal and this change might not have been documented in tIEHR by the provider.

> *I mean, the challenges are like, when the medications have changed verbally, I mean for example, the patient was taking 40 milligrams of Lasix, he or she called the PCP and said 'hat I'm feeling dizzy. The PCP told patient to cut down the dose to the half, but it was never prescribed or never documented somewhere, so in the papers, the patient is still taking 40 milligrams of Lasix, so I think that is the one issue.* (Physician, P34)

> *We have had occasion where the doctor or provider has changed dosage, usually change of dosage not change of medication, via MyGeisinger or a phone call, usually MyGeisinger, and that'doesn't get changed in the record. . ."* (Patient 4)

Communication between the inpatient and outpatient setting was noted by staff participants as especially challenging and a source of many discrepancies on patient's medication lists. Participants noted that improving communication across transitions of care would improve the process of medication reconciliation.

> *I think if you could figure out a way to communicate things, have a standardized communication, and there's that transparency between what's happening with other providers or the specialty side or what's happening from a discharge perspective, I think the communication piece, if we can fix that, that's going to be huge.* (Nurse/Operations, P37)

*Well, sometime' they're not fully updated because it may depend on the last time I saw that specialist and sometime' they're not entirely accurate, so I think if you went to all my different specialists you would see some slight disparity.* (Patient 8)

Overall, this lack of standardization led to a lack of clarity around who was responsible for medication reconciliation and varying levels of comfort with medication reconciliation.

## Lack of clarity around responsibility for medication reconciliation

While the majority of staff survey respondents (81%) agreed or strongly agreed that they had a well-defined role and knew what they were responsible for with medication reconciliation (Table 5), this differed between prescribers and non-prescribers (70% vs. 89%, respectively RRR = 0.78, 95% CI = 0.72 to 0.86, p<0.0001) and those stakeholders that were interviewed indicated that this was a potential contributor to the lack of consistency in completing medication reconciliation and may result in inaccurate medication lists.

*So, I think for us on the outpatient side, a big one is provider versus nurse, so, you know, I always feel like the final MedRec piece should be. . .it's really a provider. . .they're the ones who ultimately say yes or no this patient is not on this. A lot of times what happens is our providers just click mark as reviewed and they're not really reviewing the medications with the patient, and so, the nurses are left reconciling as best as they can and they're just removing things because the patient says I'm not taking that. . .and I don't know that that is necessarily a nursing responsibility.* (Nurse/Operations, P37)

## Comfort with medication reconciliation

Eighty-five percent of staff survey respondents felt comfortable or very comfortable with their role in medication reconciliation; non-prescribers were more likely to be comfortable with their role than prescribers (91% vs. 76%, RRR = 1.19, 95% CI = 1.10 to 1.28, p<0.0001). Level of comfort decreased when specifically asking about adding (81% vs. 85% overall) or removing (78% vs. 85% overall) medications (Table 5). Non-prescribers were more comfortable adding medications to patients' medication lists compared to prescribers (87% vs. 72%, RRR = 1.20, 95% CI = 1.10 to 1.30, p<0.0001), but non-prescribers were less comfortable removing medications compared to prescribers (75% vs. 82%, RRR = 0.92, 95% CI = 0.85 to 0.998, p = 0.04). These results may reflect unclear guidelines for removing medications from patient's medications lists (endorsed by 34% of staff respondents, 37% for non-prescribers and 29% for prescribers, RRR = 1.26, 95% CI = 0.99 to 1.59, p = 0.05) or discomfort in removing medications prescribed by others or outside of a staff members' area of expertise (endorsed by 32% of staff respondents, 26% for non-prescribers and 41% for prescribers, RRR = 0.63, 95% CI = 0.51 to 0.80, p = 0.0001) (Table 5). As interview participants noted:

*I think it's also the team, the nurse, the whoever, has a preconceived notion of what they can and cannot do or what they should and should not do. So, primary care doesn't want to discontinue a med that was ordered by a specialist. A specialist doesn't want to discontinue a med that was ordered by primary care. Even if the patient says I'm no longer taken it. The nurse. . .some nurses don't feel comfortable changing or discontinuing any medication.* (Pharmacist/Pharmacy leadership, P28)

One initiative the system put in place that may address the lack of comfort felt by nurses allows the nurses to flag medications for removal rather than having the nurse discontinue the medication themselves.

> *. . .the nurses will be able to flag a medicine rather than just discontinue it um but flag that a patient is not taking it. So, that may make it so that the nurses will be more comfortable. I know I have some nurses in my office that aren't comfortable saying that a medicine isn't being taken. Uh and so they just leave it on the list and hope that I catch it, so that new tool may enable them to feel like they can safely do that.* (Physician/Clinic Leadership, P1)

In addition to staff discomfort with medication reconciliation, patients may not be comfortable discussing certain medications (e.g. herbal supplements) or medication related issues (e.g. affordability) or feel it is unnecessary to discuss such issues with their clinician. For example, while 84% always tell their provider about their use of prescription medications, only 60% always discuss their use of herbal supplements ([Table 6]).

> *Patients may say they're taking a medication, but they're really not and they might be embarrassed to say, maybe they couldn't afford it, maybe there's other reasons but um when you hear a patient should be taking a medication everyday but they've only refilled it for a couple months supply over the course of the last year then we know it's not accurate.* (Physician, P6)

> *I guess it was because I was taking those vitamins, and I felt guilty taking them and not telling her because I didn't know if there would be an interaction between the medication and the vitamins. So, I made myself a nervous wreck.* (Patient 1)

**Knowledge.** Contributing to the lack of standardization and related to the varying degrees of comfort with medication reconciliation was a lack of knowledge about either the process of medication reconciliation or medications in general.

## Medication reconciliation process

Staff indicated that some team members may not be knowledge about the proper approach to conducting medication reconciliation.

> *I noticed with a lot of the newer nurses, they will say you know I'm going to go through your med list and the first thing a lot of the patients will say is nothing's changed. So, the new girls went to put down you know they reviewed everything, and I said to them, don't ever, ever do that because nine out of ten times, that med list is not right, and the girls started doing it and they said oh my god you are right.* (Nurse, P10)

Addressing this lack of knowledge about the process of medication reconciliation is often informal as illustrated by the quote above and by our survey which found that 70% of staff respondents reported never receiving formal medication reconciliation training from Geisinger; 37% agreed or strongly agreed that additional training would be helpful ([Table 5]). Non-prescribers were not significantly more likely to request additional training than prescribers (39% vs. 35%, RRR = 1.11, 95% CI = 0.90 to 1.38, p = 0.32). This lack of standardized training may contribute to the inconsistent performance of medication reconciliation across the organization.

**Table 6. Patient survey responses.**

| Question | Responses | Total |
|---|---|---|
| | | **(n = 577)** |
| How important is it to you that you are knowledgeable about. . . products? | Not Important | 4 (1%) |
| | Somewhat Important | 29 (5%) |
| | Important | 151 (26%) |
| | Very Important | 393 (68%) |
| I am confident I know what all of the products. . . are for. | Strongly Disagree | 12 (2%) |
| | Disagree | 14 (2%) |
| | Neither Disagree or Agree | 27 (5%) |
| | Agree | 253 (44%) |
| | Strongly Agree | 271 (47%) |
| I can describe how to use or take all of the products. . .. | Strongly Disagree | 9 (2%) |
| | Disagree | 9 (2%) |
| | Neither Disagree or Agree | 28 (4%) |
| | Agree | 251 (44%) |
| | Strongly Agree | 280 (49%) |
| How confident are you that you are able to tell a healthcare provider what products you take. . ..? | Not Confident | 8 (1%) |
| | Somewhat Confident | 56 (10%) |
| | Confident | 190 (33%) |
| | Very Confident | 323 (56%) |
| Do you currently have an up-to-date list of what products you take. . .? | Yes | 530 (92%) |
| | No | 47 (8%) |
| How confident are you that your provider has a complete. . . list of what products you take. . .? | Not Confident | 13 (2%) |
| | Somewhat Confident | 37 (6%) |
| | Confident | 170 (29%) |
| | Very Confident | 357 (62%) |
| How important is it to you that your provider knows about EVERYTHING you take. . .? | Not Important | 1 (<1%) |
| | Somewhat Important | 22 (4%) |
| | Important | 118 (20%) |
| | Very Important | 436 (76%) |

| Importance of medication type patients were taking. | | | | |
|---|---|---|---|---|
| How important is it that your provider knows about your use of. . . | Not Important | Somewhat Important | Important | Very Important |
| . . .prescriptions? (n = 563) | 2 (<1%) | 3 (<1%) | 91 (16%) | 467 (83%) |
| . . .OTC meds? (n = 325) | 1 (<1%) | 20 (6%) | 106 (33%) | 198 (61%) |
| . . .vitamins? (n = 424) | 11 (3%) | 47 (11%) | 136 (32%) | 230 (54%) |
| . . .herbal supplements? (n = 89) | 4 (4%) | 10 (11%) | 24 (27%) | 51 (57%) |
| . . . dietary supplements? (n = 88) | 2 (2%) | 12 (14%) | 29 (33%) | 45 (51%) |
| . . . other supplements? (n = 55) | 4 (7%) | 8 (15%) | 17 (31%) | 26 (47%) |
| . . . skin products? (n = 241) | 22 (9%) | 43 (18%) | 61 (25%) | 115 (48%) |
| . . .eye/nose/ear products? (n = 244) | 11 (5%) | 35 (14%) | 67 (27%) | 131 (54%) |

| Frequency of informing by medication type patients were taking. | | | | | |
|---|---|---|---|---|---|
| How often do you inform your provider about your use of. . . | Never | Rarely | About half the time | Most of the time | Always |
| . . .prescriptions? (n = 563) | 5 (1%) | 9 (2%) | 12 (2%) | 64 (11%) | 473 (84%) |

(*Continued*)

| | | | | | |
|---|---|---|---|---|---|
| ...OTC meds? (n = 325) | 5 (2%) | 21 (6%) | 19 (6%) | 69 (21%) | 211 (65%) |
| ...vitamins? (n = 424) | 17 (4%) | 26 (6%) | 23 (5%) | 67 (16%) | 291 (69%) |
| ...herbal supplements? (n = 89) | 6 (7%) | 9 (10%) | 8 (9%) | 13 (15%) | 53 (60%) |
| ...dietary supplements? (n = 88) | 4 (5%) | 9 (10%) | 4 (5%) | 17 (19%) | 54 (61%) |
| ...other supplements? (n = 55) | 4 (7%) | 5 (9%) | 3 (5%) | 7 (13%) | 36 (65%) |
| ...skin products? (n = 241) | 24 (10%) | 36 (15%) | 16 (7%) | 36 (15%) | 129 (54%) |
| ...eye/nose/ear products? (n = 244) | 17 (7%) | 29 (12%) | 20 (8%) | 43 (18%) | 135 (55%) |

| Patients bringing products to their visit | | |
|---|---|---|
| How often do bring products with you to visits? | Never | 365 (63%) |
| | Rarely | 159 (28%) |
| | About half the time | 16 (3%) |
| | Most of the time | 23 (4%) |
| | Always | 14 (2%) |
| If "Never", why not? (check all that apply) (n = 365) | No one has ever told me I should. | 212 (58%) |
| | I do not want to. | 26 (7%) |
| | I don't think it's important. | 52 (14%) |
| | I forget. | 4 (1%) |
| | I am afraid of losing them. | 1 (<1%) |
| | I am afraid of having them taken. | 1 (<1%) |
| | Other | 69 (19%) |
| How comfortable would you be bringing products in future? | Not Comfortable | 70 (12%) |
| | Somewhat Comfortable | 91 (16%) |
| | Comfortable | 215 (37%) |
| | Very Comfortable | 201 (35%) |

> *Well I think my biggest thing is, I think that there are so many different ways that people do MedRecs, that I think if there was a specific standardized course that everybody's approach to MedRec would be uniform. It would make for a lot more safer med reconciliations.* (Nurse, P44)

## Medication knowledge

In addition to lack of knowledge around how to conduct medication reconciliation, both staff and patients had varying levels of knowledge about the medications themselves which impact their ability to effectively participate in medication reconciliation. Only a minority (5%) of staff endorsed their limited amount of knowledge around medications as a potential barrier to medication reconciliation (Table 5); this did not differ by role (6% for non-prescribers vs. 4% for prescribers, RRR = 1.35, 95% CI = 0.66 to 2.78, p = 0.41). Yet, this was noted in several staff interviews as a potential reason for the lack of consistency in conducting medication reconciliation.

> *The first and foremost biggest barrier to it is the fact that in many cases, especially in the ambulatory environment, the person who is responsible for gathering the medication history*

*data has little, if any, knowledge of medications. So, that creates a great degree of uncomfortability with making any changes or alterations to the existing information. . .* (Pharmacist/ Informatician, P24)

While staff did not feel that their knowledge of medications impacted their ability to conduct medication reconciliation, staff who were interviewed consistently noted that patients are often not knowledgeable about their medications and this was a barrier to medication reconciliation. This barrier was endorsed as affecting the ability to conduct medication reconciliation at least half the time by 89% of staff survey respondents (Table 5); prescribers endorsed this barrier slightly more often than non-prescribers (91% vs. 86%, RRR = 1.07, 95% CI = 1.01 to 1.13, p = 0.03). The lack of knowledge from patients about their medications could stem from multiple causes including memory problems/cognitive impairment, illiteracy, language barriers, and having their medications managed by others.

*Well it makes it harder if um they have a poor memory or if they're not actually the one setting the medicine out. You know they depend on their spouse for their medicine when they take their medicine and that sort of thing. Like if they're not personally responsible for it.* (Physician/Clinic Leadership, P1)

Staff interviewees suggested that improved education would improve patient knowledge of their medication regimen thereby facilitating medication reconciliation. This could occur through medication lists, discharge instructions, and other educational materials which are patient friendly, written in clear and simple language, and available in the languages spoken by our patient population.

While medication lists were offered as a potential solution to improve medication reconciliation and 92% of patients reported having an up-to-date list of their medications (Table 6), not all patients bring these lists to their visits. Fifty-eight percent of staff stated that patients bring medication lists at least half the time (Table 5); non-prescribers were more likely than prescribers to report patients brought their lists in at least half the time (68% vs. 45%, RRR = 1.50, 95% CI = 1.29 to 1.75, p<0.0001). Some staff were cautious, however, about relying on patient's medication lists due to potential inaccuracies.

*. . .our medical director here, he's a big proponent of not even going by a list because patients can obviously print their med list off MyGeisinger, but it doesn't mean that that's what they have in their home, so our medical director actually, a lot of the time, will want his patients to bring in their actual bottles, so that we can match that up with actually what they're taking.* (Nurse, P44)

As highlighted in the above quote, and similar to medication lists, medication bottles can facilitate medication reconciliation, but patients rarely bring their medication bottles to the visit. Twenty percent of staff, 25% of non-prescribers and 14% of prescribers (RRR = 1.83, 95% CI = 1.28 to 2.62, p = 0.001), reported that patients bring in their bottles at least half the time which is higher than the 9% of patients who self-reported bringing their medication bottles to visits half of the time or more (Tables 5 and 6). A majority (88%) of staff noted that having the bottles at the visit would help them conduct medication reconciliation (Table 5); this attitude did not differ between non-prescribers and prescribers (87% vs. 88%, RRR = 0.99, 95% CI = 0.93 to 1.05, p = 0.79).

*I can't tell you how many times I've seen patients where I said bring in the bottle, and they bring in all their bottles and two medicines are missing, and they have no idea when that*

*happened. . . I think if we can convince our patients to bring in their bottles every single time, it would help us tremendously.* (Physician, P39)

Despite the potential benefit of bringing medication bottles to visits, 51% of staff (both prescribers and non-prescribers) reported they do not ask patients to bring in their medication bottles (Table 5). Consistent with this data, 63% of patients said they never bring their medications to visits, with 58% of those patients giving the rationale that no one had ever told them to do so (Table 6). Encouragingly, 72% of patients would either be comfortable or very comfortable bringing in their medications if asked (Table 6). Some patients felt that it was unnecessary, however, and described bringing in their medications as burdensome.

*. . .for me to do that would mean I'd have to fill a bag. I take about eight different medications every day, and I've got them organized in a such way in my bathroom, so that I'm going to take them. They sit in a certain order on my counter. I don't want to have to just dump them all in a bag and then take them home and reorganize them.* (Patient 8)

To facilitate patients bringing in their medication bottles, staff interviewees suggested implementing several initiatives including: providing patients with bags to bring their medications in, posting flyers in clinics informing patients of the importance of bringing their medications in and reminding them to do so, and mailed or telephonic reminders to bring in their medications to their next visit.

**Importance of medication reconciliation.**   Overall, both patients and staff felt that having an accurate medication list was important. Patients felt that their provider should know about everything they take, and staff felt that medication reconciliation was important or very important (96% for patients and staff) (Tables 5 and 6).

Despite the overwhelming majority of patients reporting that it was important that the provider should know about everything they take, in interviews staff expressed that a barrier to medication reconciliation was that some patients did not think medication reconciliation was important.

*. . .where the frustration comes in with the people that don't know what they're on or the people that don't pay attention or the people that just don't care. They just don't think it's that important. . .* (Nurse, P10)

This attitude was noted as a barrier to medication reconciliation at least half the time by 33% of staff respondents (Table 5); prescribers and non-prescribers endorsed this barrier at similar rates (36% vs. 29%, RRR = 1.42, 95% CI = 1.11 to 1.81, p = 0.005). Staff also reported that patients, especially those with multiple visits, get frustrated being asked about their medications repeatedly and suggested better education for patients about why it is necessary to ask about medications at each visit.

For patients, however, the issue of importance was more nuanced and varied based on the type of medical product they were taking. For example, among those taking prescriptions, 99% felt it was important or very important that the provider knows about them, while 94% and 84% felt the same for OTC medications and herbal supplements, respectively (Table 6). Further, while 84% of patients always told their provider about their use of prescription medications, only 65% and 60% always told their provider about their use of OTC and herbal medications, respectively (Table 6). This may have been due to lack of perceived harm from OTC medication.

*I think it's helpful for them to know so they have a well-rounded idea of what I'm taking over-the-counter. . . my sense is that over-the-counter medications are less likely to have adverse interactions.* (Patient 8)

The variability in the importance of medication reconciliation was also evident among staff. For example, while many staff recognized the importance of non-prescription medications, they did not consistently ask about them (OTCs were only asked about in 36% of observed visits, Table 4) due to barriers such as lack of time, lack of knowledge or appreciation of the impact of non-prescription medications, or difficulties entering these medications into the EHR.

*Nurses are rushed and I think the perceived amount of damage that can be done by over the counter medications as compared to prescription medications is thought to be much less by a hurried nurse.* (Physician, P25)

To facilitate the adherence to best practices regarding OTC medications during medication reconciliation, interviewees suggested either creating a separate section for them in the EHR or creating an alert to remind staff to specifically ask about them.

**Inadequate integration into clinical workflows.** As exemplified by the above example of OTC medications, the process of medication reconciliation is not well integrated into existing clinical workflows. In addition to difficulties associated with the EHR, staff noted that they often did not have adequate time to conduct a proper medication reconciliation.

## The electronic health record

As indicated above, entering OTC medications into the EHR was noted as challenging by some staff with 24% of staff survey respondents indicating difficulty entering patient reported medications into the EHR at least half the time (Table 5); prescribers were twice as likely to endorse this barrier than non-prescribers (35% vs. 17%, RRR = 2.13, 95% CI = 1.60 to 2.85, p<0.0001). Additional challenges to using the EHR to facilitate medication reconciliation included lack of awareness or knowledge on how to use advanced functionality such as the ability to view discrepancies between the list in the EHR and those available in claims data.

*I would tell you. I've seen that, and I am embarrassed to say that I am scared of what that is going to do. I don't know what kind of screen it will open up. I don't know what I am reconciling. Am I reconciling something from a hospitalization within Geisinger or within Care Everywhere where the meds are now flowing into the chart? Am I going to discontinue something that's connected to a care plan? I don't feel like I have enough knowledge to understand what goes on there with that button.* (Physician/Clinic Leadership, P11)

In addition, this data was often perceived to have low utility due to inaccuracies, duplications, missing data (e.g. not containing the dose of the medication), or the low relevance of the captured medications.

*I have clicked on it, and the times that I've clicked on it I've seen stuff that didn't seem to really matter much to me I guess, if I can be honest there. . .So, I just haven't really found to be enlightened by it when I used it.* (Physician Assistant, P40)

## Time

The average duration of the medication history during observations was two minutes (SD 2 minutes). Yet, nearly all staff interviewed (96%) mentioned time as a factor which affected the consistency of medication reconciliation, and 44% of staff survey respondents indicated that time influenced their ability to conduct medication reconciliation at least half the time (Table 5); prescribers were over twice as likely to endorse this barrier compared to non-prescribers (67% vs. 26%, RRR = 2.60, 95% CI = 2.13 to 3.16, p<0.0001). Staff also noted that while medication reconciliation was a key component for providing good patient care, it could be time consuming, especially for patients with many medications or those not knowledgeable about their regimen, but also given the limited amount of time to room and see patients, other priorities or pressures sometimes took precedence.

> *I think time is a huge factor. These nurses already have to do so much stuff during that rooming period that they may not necessarily have to time to do a proper MedRec.* (Physician, P45)

> *MedRecs would be updated and properly done if you were given the amount of time to do them properly...* (Nurse, P23)

To address this barrier, staff suggested allocating more time for patient visits, as well as having a separate visit to conduct a medication reconciliation (especially for new patients or for transitions of care).

## Discussion

Within a primary care environment in an integrated healthcare system, we found that inconsistency was a major factor contributing to incomplete or inaccurate medication reconciliation. This was evidenced in the inconsistent and sub-optimal adherence to best practices as well as the many identified barriers to medication reconciliation which included lack of standardized workflow, lack of knowledge about medications and the process of medication reconciliation, the variable importance of medication reconciliation (e.g., the relative lack of importance placed on OTC medications), and the inadequate integration of medication reconciliation into clinical workflows including a consistently reported lack of time to complete medication reconciliation within the current workflow. Our participants were often aware of the shortcomings of the existing medication reconciliation process and offered numerous suggestions to address the identified limitations including: education and training (for both staff and patients), standardized workflows and guidelines, EHR re-design, and patient reminders.

While the barriers encountered were common across the staff interviewed, we found that certain barriers were more likely to be endorsed by certain groups of staff. For example, we found that prescribers were over twice as likely to endorse both time and difficulty entering medications into the EHR as barriers. While the reasons behind these differences are not completely clear, the increased endorsement of time could be related to the other tasks the prescribers have to do such as examining the patient and determining a treatment plan. Patients and staff had thematically similar barriers although the specifics for each group were different. For example, while staff may not have been comfortable conducting medication reconciliation, patients were not comfortable discussing certain medications (e.g. non-prescription medications) or certain issues (e.g. affordability) with their clinical team.

To address these barriers, staff and patients made several suggestions including improved education and training, reminders, workflow standardization and redesign, EHR redesign, and increased time or designated visits to conducted medication reconciliation. A major barrier we identified was lack of training (70% of staff reported not receiving training from the

system on medication reconciliation). While discussions about medication reconciliation often occur in the didactic and experiential training of health professionals, training as part of onboarding not only emphasizes its importance and reinforces best practices, but could help to clarify responsibilities, increase comfort, and standardize the workflow across the system. Based on this potential, we are currently implementing and evaluating a training program for medication reconciliation within our system.

Our findings align with previous work examining the process of medication reconciliation in ambulatory care settings [14, 15, 21]. For example, Heyworth et al. in semi-structured interviews with healthcare providers also found that while providers recognized the importance of medication reconciliation, they also noted barriers such as the lack of a standardized process and training, patients' lack of knowledge, lack of time, and the challenges associated with communication between sites of care and utilizing the EHR for medication reconciliation [14]. Our study is unique, however, as it combines both patient and healthcare provider perspectives as well as utilizing and triangulating across multiple methodologies. Additionally, our finding of poor reconciliation of non-prescription medications aligns with findings from an inpatient study which found that among hospitalized patients, only 20% were asked about dietary supplement use and 90% disclosed that to their physician [24]. This level of disclosure is higher than what we observed, but that may be due to the difference in setting (inpatient vs. primary care). Our finding of a lack of clarity around the responsibility for medication reconciliation aligns with findings from a focus group study of inpatient nurses, physicians, and pharmacists which found that while all groups felt medication reconciliation was important, physicians felt it was not their responsibility while nurses and pharmacists felt it was the physician's responsibility [25]. This highlights the need for a clear differentiation of roles and responsibilities to ensure accurate completion of medication reconciliation regardless of the setting.

Our study leveraged mixed methods to create a detailed description of the process of medication reconciliation within a primary care environment. We observed the process across several different primary care clinics as well as interviewed and surveyed a variety of stakeholders, including nurses, physicians (both primary care and specialists), pharmacists, and patients with differing attitudes towards aspects of the medication reconciliation process. The findings across these methods were generally consistent increasing their credibility. During the observations, staff chose whether to participate and were not blind to the purpose of the study. Yet, despite this, rates of adherence to best practices were moderate (53%) indicating that these potentially optimistic estimates are below ideal (e.g. 100%) rates. In the interviews, we believe we achieved data saturation; the interviews, however, were voluntary and were limited to staff and patients at a single healthcare system potentially limiting the transferability of these findings to other healthcare systems or other patient demographics (e.g. our patient population is mostly Caucasian, including those interviewed).

We had low response rates for both our patient (11%) and staff (24%) surveys. We conducted a non-response analysis for the patient survey and found that there were some significant differences between respondents and non-respondents. Therefore, generalizing our results to all primary care patients should be done with caution. For example, younger, healthier patients with less medication use may have different attitudes toward the medication reconciliation process due to less salience. Future studies should further examine these potential differences as we did not have enough patients in either the interview or survey to robustly examine the impact of these factors on patient perception of or barriers to medication reconciliation.

While we were unable to do a non-response analysis for the staff survey, the results were consistent with our observations and interviews. To explore potential areas of variability, we compared responses between prescribers and non-prescribers and found that the prevalence

of certain barriers differs between the two groups. Some of these findings were not surprising given the different roles, but others such as prescribers more likely to endorse time as a barrier to medication reconciliation were more surprising in light of the interview data which emphasized the many tasks nurses have to complete as part of the rooming process.

There are many potential interventions which could be developed based on our findings including: educational initiatives for patients and staff to ensure both an understanding of medication as well as the process of medication reconciliation; redesign of the EHR to improve the usability both in entering medications and integrating with other data sources; and alternative technologies and workflows which free up time for staff to conduct medication reconciliation. Many studies have evaluated these or similar interventions, but while some interventions have shown effectiveness, the overall body of literature is inconsistent and engenders low confidence [6, 7, 26, 27]. To generate evidence that is likely to be effective, we can target future intervention development and deployment based on system specific barriers to medication reconciliation. These interventions will not only need to be rigorously evaluated (e.g. in well designed and conducted randomized controlled trials), but as part of the design and implementation of these interventions developers should be mindful of factors which may affect the effectiveness of such interventions, such as the acceptability and adoption of proposed interventions [28]. Frameworks such as the Consolidated Framework for Implementation Science (CFIR) identify determinates of implementation which can be useful guides in the planning of implementation efforts [29]. Future studies should be mindful of these determinants when planning and implementing evidence-based practices.

## Conclusion

An accurate medication list is essential for therapeutic decision making, yet, patients' medication lists are often inaccurate. Using a mixed methods evaluation of the process of collecting an accurate medication list in primary care, we have identified several barriers which impact the process of medication reconciliation. Our key finding is that the process of medication reconciliation is plagued by inconsistencies which contribute to inaccurate medication lists. These inconsistencies can be broken down into several categories (standardization, knowledge, importance, and inadequate integration) which can be targets for future studies and interventions. Many possible solutions were noted by patients and staff. One example, education and training, can address several of these inconsistencies and is actively being evaluated within our system. We hope that by using challenges and potential solutions derived from our staff and patients that the findings will have increased salience and lead to sustained improvements in medication reconciliation.

## Supporting information

**S1 File.**
(DOCX)

## Acknowledgments

The authors would like to thank the leadership of the clinics for facilitating the observations, Matt Gass and Ciaran Fisher for their assistance with the survey, the Geisinger Primary Care Redesign Medication Reconciliation Workgroup for their feedback on various aspects of the project, the staff and patients who participated in the project, and Drs. Rangachari and Okere for their prior work in this area which informed survey development.

## Author Contributions

**Conceptualization:** Michael R. Gionfriddo.

**Formal analysis:** Michael R. Gionfriddo, Vanessa Duboski, Allison Middernacht, Melissa S. Kern, Jove Graham, Eric A. Wright.

**Funding acquisition:** Michael R. Gionfriddo.

**Investigation:** Michael R. Gionfriddo, Vanessa Duboski, Allison Middernacht.

**Methodology:** Michael R. Gionfriddo.

**Project administration:** Michael R. Gionfriddo, Vanessa Duboski, Melissa S. Kern.

**Writing – original draft:** Michael R. Gionfriddo.

**Writing – review & editing:** Michael R. Gionfriddo, Vanessa Duboski, Allison Middernacht, Melissa S. Kern, Jove Graham, Eric A. Wright.

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
