## [Decision Letter · Decision Letter 0]

10 May 2021

PONE-D-21-10800

A Mixed Methods Evaluation of Medication Reconciliation in the Ambulatory Care Setting

PLOS ONE

Dear Dr.Gionfriddo:

Thank you for submitting your manuscript to PLOS ONE. After careful consideration, we feel that it has merit but does not fully meet PLOS ONE’s publication criteria as it currently stands. Therefore, we invite you to submit a revised version of the manuscript that addresses the points raised during the review process.

I concur with the general tenor of the reviewer's comments.  In what I think is an attempt to be rigorous, the authors have employed a number of research methods that do not clearly link together.  This, in turn, returns a significant amount of data - and the authors appear to report back on so many data points that the paper becomes very difficult to read.  The authors should consider deciding what is nice versus what is needed in both their methods and results.

We look forward to receiving your revised manuscript.

Kind regards,

John Rovers, PharmD, MIPH

Academic Editor

PLOS ONE

Journal Requirements:

"This work was funded by the Geisinger Health Plan through the Geisinger Clinic Quality Pilot Fund program for fiscal years 2020 and 2021.  The funders had no role in study design, data collection and analysis, decision to publish, or preparation of the manuscript."

Reviewers' comments:

Reviewer's Responses to Questions

**Comments to the Author**

1. Is the manuscript technically sound, and do the data support the conclusions?

Reviewer #1: Partly

Reviewer #2: Partly

2. Has the statistical analysis been performed appropriately and rigorously? 

Reviewer #1: No

Reviewer #2: No

3. Have the authors made all data underlying the findings in their manuscript fully available?

Reviewer #1: No

Reviewer #2: Yes

4. Is the manuscript presented in an intelligible fashion and written in standard English?

Reviewer #1: No

Reviewer #2: Yes

5. Review Comments to the Author

Reviewer #1: Thank you for the opportunity to review this manuscript. Upon a full reading, I do not believe that this paper does justice to the excellent opportunity this study had to collect comprehensive mixed-methods data on the topic of medication reconciliation in the outpatient setting; nor does it do justice to analyzing and presenting the mixed-method data collected.

I see several opportunities for major revision that could serve to strengthen the readability and impact of this manuscript.

• A major concern is that this paper does not provide the BIG PICTURE on study findings to address its RESEARCH QUESTIONS (and the latter in turn are not clearly stated in the INTRODUCTION). It appears that the main objective is to understand barriers to effective medication reconciliation in the outpatient setting, with special attention to barriers to obtaining an accurate medication history upon patient arrival to the clinic.

• There is no alignment across manuscript sections to address the study aims/research questions.

• A logical beginning to addressing such a research question, would be to describe the EXISTING LITERATURE on influencing factors or barriers/facilitators to effective medication reconciliation and identify GAPS that the paper would be addressing. For example, there may be a gap in understanding the med history process from the perspective of both providers and patients.

• Next, the METHODS that will be used to address the research questions should be clearly described. Presently, all three methods appear to be disparate in addressing different objectives. Data collected on adherence to best practices during observation does not appear to feed into interview or survey data collection in any meaningful way that also ties back to both the existing literature and research questions. Likewise, findings from survey are not integrated into findings from interviews in any meaningful way.

• If the authors feel that they did not have a representative sample of patients from survey, then survey data from patients should not be included. Likewise, the patient sample interviewed appears to be too small to make meaningful distinctions between patients with fewer than 5 meds, patients with 5-10 meds and patients with greater than 10 meds, which would be very important to know from perspective of accurate medication history. It also does not help to distinguish between patients by level of empowerment or health literacy which would be very relevant to the problem being examined. The section on ‘patient-related barriers,’ page 27 appears to have been developed almost entirely from staff input. If the investigators feel that the interviewed patient sample is not representative, this component should not be included in the study.

• Presently, there is no explanation of how each of the three “sequential” methods complemented each other to provide a holistic understanding of the problem of interest to address the research question. I would recommend reviewing the book on mixed method research design by Creswell and Creswell (2018) for tips on presenting results from mixed-methods research.

• There are also multiple issues with the presentation of RESULTS.

• The presentation of qualitative results begins abruptly on pages 10 and 11 with no preparation for the reader. There are no subheadings related to thematic categories on these pages and then we see the first category on ‘staff-related’ barriers introduced on page 12. Frankly, the thematic categories of staff and patient related barriers appear to be weak, and not impactful in providing insight into strategies for overcoming barriers identified to accurate medication history.

• Moreover, the categorization of ‘technological barriers’ under ‘logistical problems’ does not appear to do justice to the critical problem which the write-up appears to be depicting, i.e., a MISMATCH BETWEEN CLINICAL WORKFLOW AND EHR SYSTEM DESIGN, which in turn can provide insight into impactful strategies for process and system redesign for med history process on EHR.

• Following form the above concern, there needs to be more transparency related to the coding methodology used for qualitative (thematic) analysis. How were these categories identified, and how do they tie back to the existing literature on barriers already identified, and how can they be leveraged to gain insight into strategies for improvement?

• Overall, the paper gets caught up in the DETAILS of the qualitative study with a preponderant focus on presenting detailed comments from staff and patient interviews (which is frankly fatiguing to the reader, since it is a mishmash of comments that could belong in any of the thematic categories specified). If the paper is going to focus primarily on qualitative data obtained from staff interviews, then I would recommend a full revision of the presentation of results to focus on the BIG PICTURE findings related to barriers to effective med history, with a supporting table on DETAILS from qualitative comments.

• In a similar vein, the DISCUSSION section references interventions that have been implemented in the literature, however, the discussion needs to provide a clear summary of findings, discuss limitations and then outline a clear set of implications from the study for practice and future research, including insights into strategies and interventions for overcoming barriers identified to med history identified in the study. Likewise, the CONCLUSION needs to be strengthened to more impactful in conveying the overall contribution of the study to the literature and to addressing the problem of interest.

In summary, there are many substantial insights that could emanate from this study, on both barriers to med history and strategies for overcoming them, but it will require a major revision of this manuscript to flesh these out and realize the full potential of the study.

Thank you for the opportunity to review this manuscript.

Reviewer #2: This research is a very useful contribution to an important area of clinical practice, thank you. Investigation of medication reconciliation practices and attitudes and behaviors in primary care settings is identified as a gap in the literature.

A mixed methods approach to describe and explore this topic is appropriate, and may be useful to generate hypotheses for future research.

My concerns with this manuscript are primarily focused on the way in which the study is described and presented.

1. The stated aim of the study is:

'To understand the extent to which behaviors consistent with high quality medication

reconciliation occurred in primary care settings and which factors contribute to

inaccurate medication lists', however that aim does not appear to be achieved with the study as described.

The work that has been presented does explore practices/behaviors and attitudes, but is not designed to '...understand the extent to which behaviors consistent with .... contribute to inaccurate medication lists.'

I suggest that the study objective be reworded to align with the study methods as presented in the paper.

2. In the first sentence of the paper (Line 44) "Failure to conduct proper medication reconciliation results in....' - the use of the word 'proper' is unusual in a scientific paper. While this may be conversational and understood, it would be more appropriate to consider other terms. For example, does this mean- inaccurate, undocumented, incomplete, absent…?

3. Definition of medication reconciliation. The reader would expect to see the precise definition of the term, as used by the authors in this research given the nature of the study. The literature has used this term widely. Furthermore the audience for this journal may not have a single interpretation. In the manuscript it is stated: (line 45) ‘Medication reconciliation is the process of creating an accurate medication list'. This might be considered a limited definition, given the work in the manuscript.

The authors have referenced a paper (#5) that does provide a comprehensive statement:

"... a proposed definition for medication reconciliation tasks as "the process of creating the most accurate list possible of all medications a patient is taking and comparing that list against the prescriber's orders. In addition, the patient's allergies, history of side effects from medications and medication aids are listed with the goal of providing correct medication to the patient at all transition points within the health care system."

It would seem key to include a definitive statement of medication reconciliation in the introduction to this paper.

4. The sentence in lines 70-72 appear incomplete, a typographical error.

"The observations were considered quality improvement (IRB#2019-0561) while consent was obtained for the surveys and interviews and deemed exempt (IRB# 2019-0868)"

5. Line 79: 'rooming staff' – this term may need definition for an international audience.

6. Line 164: "All statistical analysis was done with SAS." Rather than identifying the software product, please outline the statistical tests/methods/analyses that were utilized to address the specific quantitative research aims.

7. Line 665: "... a thorough mixed methods evaluation" The word ‘thorough’ is not needed, nor appropriate. If the authors intend to highlight the rigorous or comprehensive nature of their study, then that should be stated and justified. It appears that the intention here is to declare that this study is appropriately designed and conducted in such a way to meet the study objectives.

6. PLOS authors have the option to publish the peer review history of their article (what does this mean?). If published, this will include your full peer review and any attached files.

Reviewer #1: No

Reviewer #2: No

---

## [Author Response · Author response to Decision Letter 0]

14 Jun 2021

On behalf of the authors, I would like to thank the editor and reviewers for their suggestions. We have attempted to address these suggestions through a significant re-write and re-organization of the manuscript. Please find below a point-by-point response to the concerns that were raised. 

Sincerely,

Michael Gionfriddo

Comments from the Academic Editor

I concur with the general tenor of the reviewer's comments. In what I think is an attempt to be rigorous, the authors have employed a number of research methods that do not clearly link together. This, in turn, returns a significant amount of data - and the authors appear to report back on so many data points that the paper becomes very difficult to read. The authors should consider deciding what is nice versus what is needed in both their methods and results.

Response

After careful consideration we have completely re-organized the results section and in the process significantly shortened the manuscript. We hope this has added the clarity and conciseness sought by the editor. We however, are concerned about shortening the methods section as we believe they are important for readers to understand any potential biases or limitations in our approach.

Comments from Reviewer 1

Comment 1

Thank you for the opportunity to review this manuscript. Upon a full reading, I do not believe that this paper does justice to the excellent opportunity this study had to collect comprehensive mixed-methods data on the topic of medication reconciliation in the outpatient setting; nor does it do justice to analyzing and presenting the mixed-method data collected.

I see several opportunities for major revision that could serve to strengthen the readability and impact of this manuscript.

• A major concern is that this paper does not provide the BIG PICTURE on study findings to address its RESEARCH QUESTIONS (and the latter in turn are not clearly stated in the INTRODUCTION). It appears that the main objective is to understand barriers to effective medication reconciliation in the outpatient setting, with special attention to barriers to obtaining an accurate medication history upon patient arrival to the clinic. 

• There is no alignment across manuscript sections to address the study aims/research questions.

Response

We have updated the objective based on reviewers’ feedback. The objective now reads: To understand the extent to which behaviors consistent with high quality medication reconciliation occurred in primary care settings and explore barriers to high quality medication reconciliation. We have also added the specific research questions the study aimed to answer which were:

1. To what extent are behaviors indicative of a best possible medication history for medication reconciliation adhered to in a primary care setting?

2. What are the barriers which inhibit the accurate completion of medication reconciliation? Do these barriers differ between patients and staff or between staff with and without prescribing authority?

3. What changes do patients and staff think can be made to improve the process of medication reconciliation in primary care? 

At the beginning of the discussion we tie our findings back to these research questions to hopefully improve upon the lack of alignment noted by the reviewer. 

Comment 2

• A logical beginning to addressing such a research question, would be to describe the EXISTING LITERATURE on influencing factors or barriers/facilitators to effective medication reconciliation and identify GAPS that the paper would be addressing. For example, there may be a gap in understanding the med history process from the perspective of both providers and patients.

Response

We have modified the introduction to better highlight the gaps in the existing literature.

Comment 3

• Next, the METHODS that will be used to address the research questions should be clearly described. Presently, all three methods appear to be disparate in addressing different objectives. Data collected on adherence to best practices during observation does not appear to feed into interview or survey data collection in any meaningful way that also ties back to both the existing literature and research questions. Likewise, findings from survey are not integrated into findings from interviews in any meaningful way.

Response

We decided to present the methods separately for clarity, however we clearly indicate how each method contributed to the development of the study design. In the results however, the findings from the interviews, surveys, and observations are triangulated to present a more complete picture of the process of and barriers to medication reconciliation. Overall, across each of the methods the findings were concurrent and reinforced each other. In response to the reviewer’s comment, we have tried to tie our findings to additional literature within the discussion. 

Comment 4

• If the authors feel that they did not have a representative sample of patients from survey, then survey data from patients should not be included. Likewise, the patient sample interviewed appears to be too small to make meaningful distinctions between patients with fewer than 5 meds, patients with 5-10 meds and patients with greater than 10 meds, which would be very important to know from perspective of accurate medication history. It also does not help to distinguish between patients by level of empowerment or health literacy which would be very relevant to the problem being examined. The section on ‘patient-related barriers,’ page 27 appears to have been developed almost entirely from staff input. If the investigators feel that the interviewed patient sample is not representative, this component should not be included in the study.

Response

As part of responding to reviewers comments, we have completely reorganized the results section including the themes and subthemes. As a result, we have removed the section on patient related barriers and integrated those into the relevant themes where appropriate. In this way the patient data is used to help triangulate the themes which were identified by the staff. We concur that further research is needed on patient barriers and perceptions of medication reconciliation and have added a statement to the discussion saying so. 

Comment 5

• Presently, there is no explanation of how each of the three “sequential” methods complemented each other to provide a holistic understanding of the problem of interest to address the research question. I would recommend reviewing the book on mixed method research design by Creswell and Creswell (2018) for tips on presenting results from mixed-methods research.

Response

We understand how it may not have been clear how our methods complemented each other. To clarify we have added an additional paragraph to the methods which reads: We conducted a fully mixed sequential equal status mixed methods[16] evaluation of medication reconciliation within ambulatory care clinics combining ethnographic observations, semi-structured interviews, and surveys. The observations occurred first and informed both survey and interview guide development. Subsequently, we performed staff interviews from which we identified themes that led us to create targeted questions to assess in the staff survey to understand the applicability of the themes to the broader population of staff at our institution. The patient survey, which was informed by the observations and interviews with staff was then used to recruit patients to interview. The findings from each method are mixed in the results based on thematic categorization and act as a form of triangulation.

Comment 6

• There are also multiple issues with the presentation of RESULTS.

• The presentation of qualitative results begins abruptly on pages 10 and 11 with no preparation for the reader. There are no subheadings related to thematic categories on these pages and then we see the first category on ‘staff-related’ barriers introduced on page 12. Frankly, the thematic categories of staff and patient related barriers appear to be weak, and not impactful in providing insight into strategies for overcoming barriers identified to accurate medication history.

Response

We have substantially modified the presentation of the results based on reviewer feedback. We have moved away from the framework of staff and patient barriers and there is now an overarching theme (Inconsistency) with introductory text to guide the reader in the presentation of the results. 

Comment 7

• Moreover, the categorization of ‘technological barriers’ under ‘logistical problems’ does not appear to do justice to the critical problem which the write-up appears to be depicting, i.e., a MISMATCH BETWEEN CLINICAL WORKFLOW AND EHR SYSTEM DESIGN, which in turn can provide insight into impactful strategies for process and system redesign for med history process on EHR.

Response

We thank the reviewer for this comment and as part of a restructuring of themes have incorporated the reviewer’s suggestion. There is now a theme around inadequate integration of medication reconciliation into clinical workflows which includes issues with the electronic health record.

Comment 8

• Following form the above concern, there needs to be more transparency related to the coding methodology used for qualitative (thematic) analysis. How were these categories identified, and how do they tie back to the existing literature on barriers already identified, and how can they be leveraged to gain insight into strategies for improvement?

Response

We have clarified in the text that the coding was inductive and that themes were collaboratively generated based on discussion with the team upon reviewing the inductively identified codes. Additionally, we have added additional discussion about how some of the themes tie back to existing literature as well as better clarifying potential areas for improvement based on our findings. 

Comment 9

• Overall, the paper gets caught up in the DETAILS of the qualitative study with a preponderant focus on presenting detailed comments from staff and patient interviews (which is frankly fatiguing to the reader, since it is a mishmash of comments that could belong in any of the thematic categories specified). If the paper is going to focus primarily on qualitative data obtained from staff interviews, then I would recommend a full revision of the presentation of results to focus on the BIG PICTURE findings related to barriers to effective med history, with a supporting table on DETAILS from qualitative comments.

Response

We have substantially revised and reorganized the manuscript to improve readability and focus including the removal or paring down of several quotes. We believe the remain quotes are more focused and help to ground the themes in the data which adds credibility to our findings. 

Comment 10

• In a similar vein, the DISCUSSION section references interventions that have been implemented in the literature, however, the discussion needs to provide a clear summary of findings, discuss limitations and then outline a clear set of implications from the study for practice and future research, including insights into strategies and interventions for overcoming barriers identified to med history identified in the study. Likewise, the CONCLUSION needs to be strengthened to more impactful in conveying the overall contribution of the study to the literature and to addressing the problem of interest.

In summary, there are many substantial insights that could emanate from this study, on both barriers to med history and strategies for overcoming them, but it will require a major revision of this manuscript to flesh these out and realize the full potential of the study.

Thank you for the opportunity to review this manuscript.

Response

In response to the reviewers comment, we have substantially revised the discussion including a new summary of the results and conclusion. 

Comments from Reviewer 2

Comment 1

This research is a very useful contribution to an important area of clinical practice, thank you. Investigation of medication reconciliation practices and attitudes and behaviors in primary care settings is identified as a gap in the literature.

A mixed methods approach to describe and explore this topic is appropriate, and may be useful to generate hypotheses for future research.

My concerns with this manuscript are primarily focused on the way in which the study is described and presented.

1. The stated aim of the study is:

'To understand the extent to which behaviors consistent with high quality medication

reconciliation occurred in primary care settings and which factors contribute to

inaccurate medication lists', however that aim does not appear to be achieved with the study as described.

The work that has been presented does explore practices/behaviors and attitudes, but is not designed to '...understand the extent to which behaviors consistent with .... contribute to inaccurate medication lists.'

I suggest that the study objective be reworded to align with the study methods as presented in the paper.

Response

Based on the reviewer’s feedback, we have revised our objective statement to more accurately reflect the aim of our study which now reads: “To understand the extent to which behaviors consistent with high quality medication reconciliation occurred in primary care settings and explore barriers to high quality medication reconciliation.”

Comment 2

2. In the first sentence of the paper (Line 44) "Failure to conduct proper medication reconciliation results in....' - the use of the word 'proper' is unusual in a scientific paper. While this may be conversational and understood, it would be more appropriate to consider other terms. For example, does this mean- inaccurate, undocumented, incomplete, absent…?

Response

We have changed this fragment to now read: “Inaccurate or incomplete medication reconciliation…”

Comment 3

3. Definition of medication reconciliation. The reader would expect to see the precise definition of the term, as used by the authors in this research given the nature of the study. The literature has used this term widely. Furthermore the audience for this journal may not have a single interpretation. In the manuscript it is stated: (line 45) ‘Medication reconciliation is the process of creating an accurate medication list'. This might be considered a limited definition, given the work in the manuscript.

The authors have referenced a paper (#5) that does provide a comprehensive statement:

"... a proposed definition for medication reconciliation tasks as "the process of creating the most accurate list possible of all medications a patient is taking and comparing that list against the prescriber's orders. In addition, the patient's allergies, history of side effects from medications and medication aids are listed with the goal of providing correct medication to the patient at all transition points within the health care system."

It would seem key to include a definitive statement of medication reconciliation in the introduction to this paper.

Response

We have now added the full definition of medication reconciliation to the manuscript.

Comment 4

4. The sentence in lines 70-72 appear incomplete, a typographical error.

"The observations were considered quality improvement (IRB#2019-0561) while consent was obtained for the surveys and interviews and deemed exempt (IRB# 2019-0868)"

Response

To improve clarity, we have now revised this sentence to read: The observations were considered quality improvement (IRB#2019-0561), while the surveys and interviews were deemed exempt (IRB# 2019-0868).

Comment 5

5. Line 79: 'rooming staff' – this term may need definition for an international audience.

Response

Thank you for this comment, we have replaced rooming staff with “nurse or medical assistant”

Comment 6

6. Line 164: "All statistical analysis was done with SAS." Rather than identifying the software product, please outline the statistical tests/methods/analyses that were utilized to address the specific quantitative research aims.

Response

We have added additional detail on the statistical analysis. The section now reads: Comparisons of continuous data were conducted using t-tests, while comparisons of categorical data were conducted using chi-square or Fisher’s exact tests as appropriate. All statistical analyses were done with SAS (SAS 9.4, Cary, NC) or R (The R Group, Vienna, Austria) and p-values <0.05 were considered significant; no adjustment was made for multiple comparisons.

Comment 7

7. Line 665: "... a thorough mixed methods evaluation" The word ‘thorough’ is not needed, nor appropriate. If the authors intend to highlight the rigorous or comprehensive nature of their study, then that should be stated and justified. It appears that the intention here is to declare that this study is appropriately designed and conducted in such a way to meet the study objectives.

Response

We have removed this adjective at the advice of the reviewer.

---

## [Decision Letter · Decision Letter 1]

13 Jul 2021

PONE-D-21-10800R1

A Mixed Methods Evaluation of Medication Reconciliation in the Primary Care Setting

PLOS ONE

Dear Dr. Gionfriddo,

Thank you for submitting your manuscript to PLOS ONE. After careful consideration, we feel that it has merit but does not fully meet PLOS ONE’s publication criteria as it currently stands. Therefore, we invite you to submit a revised version of the manuscript that addresses the points raised during the review process.

The manuscript is much improved from the first version so, thank you very much for that.  Reviewer #1 still has several suggestion (that I agree with) that will further strengthen the paper.  If you can address these adequately (and explain the changes in your rebuttal letter) I should be able to accept the paper without further review or revision.

We look forward to receiving your revised manuscript.

Kind regards,

John Rovers, PharmD, MIPH

Academic Editor

PLOS ONE

Journal Requirements:

Reviewers' comments:

Reviewer's Responses to Questions

**Comments to the Author**

1. If the authors have adequately addressed your comments raised in a previous round of review and you feel that this manuscript is now acceptable for publication, you may indicate that here to bypass the “Comments to the Author” section, enter your conflict of interest statement in the “Confidential to Editor” section, and submit your "Accept" recommendation.

Reviewer #1: (No Response)

Reviewer #2: All comments have been addressed

2. Is the manuscript technically sound, and do the data support the conclusions?

Reviewer #1: Yes

Reviewer #2: Yes

3. Has the statistical analysis been performed appropriately and rigorously? 

Reviewer #1: Yes

Reviewer #2: Yes

4. Have the authors made all data underlying the findings in their manuscript fully available?

Reviewer #1: Yes

Reviewer #2: Yes

5. Is the manuscript presented in an intelligible fashion and written in standard English?

Reviewer #1: Yes

Reviewer #2: Yes

6. Review Comments to the Author

Reviewer #1: This is a substantially improved version of the manuscript! I appreciate the authors’ high responsiveness to reviewer recommendations. I do however have some suggestions for minor revisions at this stage to improve the overall organization and flow of the manuscript, particularly in regard to strengthening the ALIGNMENT between research questions and presentation of methods, results, discussion and conclusion.

METHODOLOGY

• It would be helpful to have a brief explanation of how the observation component of the methodology helped to develop the interview guide and likewise, how both observations and interviews contributed to development of the survey. Just a few key EXAMPLES would be helpful in illustrating the complementarity among the three methods in providing a holistic understanding of the problem (and addressing the three research questions).

• It would also be helpful to clarify that Research Question 1 is addressed by observation, while Research Questions 2 and 3 are addressed through surveys and interviews.

RESULTS

• It would be helpful to have broad subsections corresponding to the three research questions in the Results to improve alignment. E.g., results related to: 1) medication reconciliation best practice behaviors; 2) barriers, and 3) suggestions for improvement.

• Line 287-288 in Track Changes version requires clarification. It says non-prescribers were more likely to agree there was a standardized process but the quotations do not corroborate this.

• It would be most helpful to have a table summarizing the key results related to research question 2 (i.e., barriers) and research question 3 (i.e., corresponding suggestions for improvement). Even a simple listing of inconsistencies related to workflow, communication, documentation etc. (from perspective of staff/patient) with corresponding simple suggestions for improvement would greatly help to strengthen the presentation of results.

• The narrative would serve to supplement such a key results table in highlighting relevant quotations.

DISCUSSION

• It would be important to elaborate on findings related to Research Question 3 (suggestions for improvement) in the Discussion (e.g., some details related to standardized workflows, education, training, EHR redesign etc.).

CONCLUSION

• It would be helpful to include a couple of sentences related to suggestions for improvement (research question 3) to provide a complete picture of the study contribution in the conclusion.

Again, I appreciate the authors’ efforts in revising this paper! Best wishes.

Reviewer #2: thank you for these editing changes. I think this is now a 'stronger' manuscript, and will be well received. Congratulations

7. PLOS authors have the option to publish the peer review history of their article (what does this mean?). If published, this will include your full peer review and any attached files.

Reviewer #1: **Yes: **Dr. Pavani Rangachari

Reviewer #2: No

---

## [Author Response · Author response to Decision Letter 1]

23 Jul 2021

We have addressed the remaining review comments. Please let us know if the manuscript requires additional edits.

---

## [Decision Letter · Decision Letter 2]

19 Nov 2021

A Mixed Methods Evaluation of Medication Reconciliation in the Primary Care Setting

PONE-D-21-10800R2

Dear Dr. Gionfriddo,

We’re pleased to inform you that your manuscript has been judged scientifically suitable for publication and will be formally accepted for publication once it meets all outstanding technical requirements.

Kind regards,

Lucinda Shen, MSc

Staff Editor

PLOS ONE

Additional Editor Comments (optional):

Reviewers' comments:

Reviewer's Responses to Questions

**Comments to the Author**

1. If the authors have adequately addressed your comments raised in a previous round of review and you feel that this manuscript is now acceptable for publication, you may indicate that here to bypass the “Comments to the Author” section, enter your conflict of interest statement in the “Confidential to Editor” section, and submit your "Accept" recommendation.

Reviewer #1: All comments have been addressed

Reviewer #2: All comments have been addressed

2. Is the manuscript technically sound, and do the data support the conclusions?

Reviewer #1: Yes

Reviewer #2: Yes

3. Has the statistical analysis been performed appropriately and rigorously? 

Reviewer #1: Yes

Reviewer #2: Yes

4. Have the authors made all data underlying the findings in their manuscript fully available?

Reviewer #1: Yes

Reviewer #2: Yes

5. Is the manuscript presented in an intelligible fashion and written in standard English?

Reviewer #1: Yes

Reviewer #2: Yes

6. Review Comments to the Author

Reviewer #1: Thank you to authors for their high responsiveness to reviewer recommendations. The latest revised version satisfactorily addresses my comments and suggestions. I have no additional revisions to recommend. My best wishes to authors in future efforts to disseminate their work and results.

Reviewer #2: This manuscript has been carefully revised (twice)- and the comments made by the reviewers have been incorporated in a thoughtful and reasonable way.

I recommend acceptance for publication

7. PLOS authors have the option to publish the peer review history of their article (what does this mean?). If published, this will include your full peer review and any attached files.

Reviewer #1: **Yes: **Dr. Pavani Rangachari

Reviewer #2: **Yes: **Prof Jo-anne Brien

---

## [Editor Report · Acceptance letter]

23 Nov 2021

PONE-D-21-10800R2 

A Mixed Methods Evaluation of Medication Reconciliation in the Primary Care Setting 

Dear Dr. Gionfriddo:

I'm pleased to inform you that your manuscript has been deemed suitable for publication in PLOS ONE. Congratulations! Your manuscript is now with our production department. 

Kind regards, 

on behalf of

Miss Lucinda Shen 

Staff Editor

PLOS ONE